# An HGA-LSTM-Based Intelligent Model for Ore Pulp Density in the Hydrometallurgical Process

**DOI:** 10.3390/ma15217586

**Published:** 2022-10-28

**Authors:** Guobin Zou, Junwu Zhou, Kang Li, Hongliang Zhao

**Affiliations:** 1College of Information Science and Engineering, Northeastern University, Shenyang 110004, China; 2State Key Laboratory of Process Automation in Mining and Metallurgy Research, Beijing 100160, China; 3BGRIMM Technology Group, Beijing 100160, China; 4School of Metallurgical and Ecological Engineering, University of Science and Technology, Beijing 100083, China

**Keywords:** intelligent model, thickening process, ore pulp density, long short-term memory, hybrid genetic algorithm

## Abstract

This study focused on the intelligent model for ore pulp density in the hydrometallurgical process. However, owing to the limitations of existing instruments and devices, the feed ore pulp density of thickener, a key hydrometallurgical equipment, cannot be accurately measured online. Therefore, aiming at the problem of accurately measuring the feed ore pulp density, we proposed a new intelligent model based on the long short-term memory (LSTM) and hybrid genetic algorithm (HGA). Specifically, the HGA refers to a novel optimization search algorithm model that can optimize the hyperparameters and improve the modeling performance of the LSTM. Finally, the proposed intelligent model was successfully applied to an actual thickener case in China. The intelligent model prediction results demonstrated that the hybrid model outperformed other models and satisfied the measurement accuracy requirements in the factory well.

## 1. Introduction

Hydrometallurgy is important in mineral resources. The hydrometallurgical process can deal with low-grade mines, complex ores, and generates fewer emissions to the environment [1]. The thickening process is a typical process of hydrometallurgy. The optimized control technology for hydrometallurgical processing is of great applicational value for the efficient utilization of metal mineral resources [2]. Optimal control of the thickening process usually depends on quality variables, such as feed density [3], which are difficult to measure online because the density of the feeding ore usually fluctuates substantially due to the existence of nonlinearity [4]. However, there is no research on the application of real-time online measurement methods of thickener feed pulp density in the actual production process [5].

To alleviate these problems, intelligent models have been used to predict wind flow around buildings by establishing inferential mathematical prediction models [6]. Owing to rapid response, accurate prediction results, and low maintenance costs, intelligent models have currently become one of the main methods for detecting quality variables in industrial processes, such as wind-induced pressure prediction [7], temperature prediction for roller kiln [8], and surface crack detection [9].

Considering the wide implementation of distributed control systems and the massive amount of available data, soft sensors based on data-driven systems are receiving increasing attention [10]. Typical data-driven modeling methods include many multivariate statistical and machine-learning methods [11,12,13]. Because of the limitations of the structures and parameters, some methods are limited to present strong nonlinearities and dynamics. In the past decade, deep learning has drawn increasing attention in many fields, such as intelligent model applications [14], image classification [15], and process monitoring [16]. Compared with existing modeling methods, deep neural networks (DNNs) have a significant ability to express complex functions and learn the primary highlights of data [17]. The DNN model has shown excellent performance in processing complex and strongly nonlinear data for the development of intelligent industrial models. Most of these existing intelligent models are deep static models based on the assumptions of steady state, such as stacked auto-encoders (SAEs) [18] and deep belief networks (DBNs) [19]. Nevertheless, industrial processes are naturally dynamic, and the data series is sampled in real-time from a continuous process. Thus, to model such data sequences more accurately, the dynamic characteristics must also be considered; that is, the models must utilize past states and information to predict the present state.

Recurrent neural networks (RNNs) are dynamic neural networks. They can suffer from gradient explosion and gradient disappearance because of the memory function of past information [20]. Thus, a long short-term memory (LSTM) network which adds gate units to retain short and long-term memories is proposed to deal with this problem [21]. Recently, Zhang set up an LSTM-based network in the zinc flotation circuit to estimate the tailings grade of the first rougher [22]. Pan proposed an intelligent model based on an LSTM network to estimate the oxygen content of boiler flue gas [23]. Wensi developed a soft sensor method based on an LSTM network structure to handle the strong dynamics and nonlinearity of the process and verified its power using a sulfur recovery unit benchmark [24].

To satisfy the measurement requirements, the intelligent model focuses on constructing an accurate estimation. However, it is worth noting that LSTM network models have numerous hyperparameters that need to be continuously modified to obtain the most suitable results, such as the time window size and network structure [25]. Selecting the best hyperparameters is essential to optimize the validation errors, but it is extremely time-consuming. Therefore, the most commonly used method in hyperparameter estimation is the trial-and-error method based on heuristics. However, the limitations of computation level and time make it impossible to traverse the entire parameter space [26]. Thus, a new method is needed to optimize the verification error for both boosting accuracy and saving time, which can ensure the accuracy of the soft sensors in industrial processes. The genetic algorithm (GA) is a classic global optimal method developed by imitating the natural biological evolution mechanism and has attracted much attention in hyperparameter optimization. Most recently, Alshwaheen proposed an LSTM-RNN model to forecast the deterioration of ICU patients and used a modified GA to optimize the observation window to increase the accuracy [27]. Danial developed both ANN and GA-based ANN techniques for the prediction of AOP [28]. Zhang et al. combined a support vector machine (SVM) with GA to predict the moisture in oil-immersed insulations and obtained highly accurate results [29].

In this research, an intelligent model method based on LSTM and the hybrid genetic algorithm (HGA) was proposed to measure the feed ore pulp density in the thickener process. In particular, we applied the sequential quadratic programming (SQP) algorithm, which can perform fast and accurate local searches in GA and significantly increase the global searching ability of the algorithm. The GA-SQP, also called HGA, is used to optimize the hyperparameters to determine the time window and the structure parameters of the LSTM network based on the lowest verification error, which can enhance the performance of the LSTM. Finally, an intelligent modeling method was applied to a real thickener in China.

The contributions of this study:We introduced an intelligent model to resolve the difficulty encountered in measuring the feed density in the thickener through online real-time detection in hydrometallurgy.A novel intelligent modeling method combining SQP, GA, and LSTM was developed to address the nonlinear and dynamic background. The HGA algorithm was used to optimize the hyperparameters of the LSTM.From the perspective of actual cases, the results show that the method fulfills the measurement requirements in the factory.

## 2. Methodology

### 2.1. LSTM Network

An RNN is a dynamic neural network with an internal connection that can utilize past information and past states for the present state estimation [30]. Figure 1 is the structure of the RNN in the time step. RNNs also have a hidden state vector or memory and generate an output. The RNN has difficulty learning long input sequences and can easily produce gradient explosion or disappearance.

LSTM, a variant of the RNN architecture, designs a unique LSTM unit that can preserve past information and past states and learn sequential information with long-term dependencies. In Figure 2, the structure of the LSTM is presented, and the LSTM forward calculation formulas are as follows [21]:(1)$$f_{t}=\sigma (W_{f}\times \left[h_{t-1},x_{t},c_{t-1}\right]+b_{f})$$
(2)$$i_{t}=\sigma (W_{i}\times \left[h_{t-1},x_{t},c_{t-1}\right]+b_{i})$$
(3)$${\hat{c}}_{t}=\tanh(W_{c}\times \left[h_{t-1},x_{t},c_{t-1}\right]+b_{c})$$
(4)$$c_{t}=f_{t}\times c_{t-1}+i_{t}\times {\hat{c}}_{t}$$
(5)$$o_{t}=\sigma (W_{o}\times \left[h_{t-1},x_{t},c_{t-1}\right]+b_{o})$$
(6)$$h_{t}=o_{t}\times \tanh\left(c_{t}\right)$$
(7)$${\hat{y}}_{t}=(W_{y}\times h_{t}+b_{y})$$
where σ and tanh represent the sigmoid and tanh activation functions, ***W*** and ***b*** represent the matrices of the weight parameter and the bias, respectively, and the subscripts “*i*”, “*f*”, “*c*”, “*o*”, and “*y*” represent the input-gate, forgetting-gate, update-gate, and output-gate, respectively.

Through the calculation method, the long-term dependence on traditional RNN training can be overcome by the LSTM architecture effectively [31].

### 2.2. Hybrid Genetic Algorithm

GA is an adaptive heuristic optimization algorithm based on a computational model simulating the natural evolution process and has been used to determine near-optimal solutions. The algorithm uses mathematics and computer simulation to transform the process of problem-solving into a process of chromosome mutation and crossover in natural biological evolution mechanisms. Compared with some traditional optimization methods, the GA can usually obtain better optimization results more quickly when solving more complex system problems. However, the local search efficiency of a typical GA is low and time-consuming. With the stage of evolution, lower search efficiency and multiple calculations are required to achieve the final convergence [32].

The SQP algorithm is an effective method for solving nonlinear optimization problems [33]. Compared with other methods, the SQP has high computational efficiency, good convergence, and strong boundary-searching ability. The nonlinear optimization problem is expressed as follows:(8)$$\left\{ \begin{array}{l} \min f(x)\\ subject\;to\;g_{i}(x)\leq 0\;(i=1,2,\ldots ,m_{p})\\ \;\;\;\;\;\;\;\;\;\;\;\;\;\;\;\;\;\;g_{i}(x)=0\;(i=m_{p}+1,\ldots ,m) \end{array}\right.$$
where f(x) represents the objective optimization function and $g_{i}(x)$ represents the boundary conditions. The subproblem is obtained by approximating the language function quadratically and linearizing the nonlinear constraints.
(9)$$L(x,\lambda )=f(x)+\sum_{i=1}^{m}\lambda_{i}g_{i}$$
where $\lambda_{i}$ is a language factor. The Hessian matrix is approximated by the quasi-Newtonian. At each $x_{k}$, the quadratic programming (QP) subproblem is obtained by linearizing the nonlinear constraints.
(10)$$\left\{ \begin{array}{l} \min\frac{1}{2}d^{T}H_{K}d+\nabla f{\left(x_{k}\right)}^{T}d\\ subject\;to\;\nabla g_{i}{\left(x_{k}\right)}^{T}+g_{i}(x_{k})\leq 0\;(i=1,2,\ldots ,m_{p})\\ \;\;\;\;\;\;\;\;\;\;\;\;\;\;\;\;\;\;\nabla g_{i}{\left(x_{k}\right)}^{T}+g_{i}(x_{k})=0\;(i=m_{p}+1,\ldots ,m) \end{array}\right.$$

$x_{k}$, $\lambda_{k}$, and $H_{k}$ are the approximations of the solution, multiplier, and Hessian of the language function, respectively.

The search direction $d_{k}$ of the current iteration can be obtained using the above formula, and an iteration point is calculated using the formula:(11)$$x_{k+1}=x_{k}+a_{k}d_{k}$$

As a result, an HGA has been proposed by integrating the SQP algorithm and GA [34]. The HGA refers to a novel optimization search algorithm model, and the SQP can perform a fast and accurate local search in the GA to significantly increase the global search ability. First, using the excellent global search ability of the GA, all solutions in the solution space can be searched quickly, and some convergence values can be obtained without falling into the trap of local optimal solutions with rapid gradient descent. Moreover, the convergent result of each iteration can be the initial value of the SQP. Subsequently, the SQP search algorithm is used to implement a powerful local search designed to pursue a global optimal solution. Briefly, the HGA perfectly unites excellent global and fast local search capabilities.

In Figure 3, the specific steps of the hybrid algorithm are listed below. First, we determined a series of convergent populations that fulfilled the constraints through the GA and selected and retained suitable individuals to solve the initial value of the SQP. Second, we constructed a multiplier function to determine whether the prediction criteria were satisfied. When the prediction criteria were not met, the vector of the local search was identified, and the minimum point in the direction was continuously determined.

## 3. Process Description

This work focused on mineral processing and attempted to solve the dilemma of the online measurement of the feed density in a thickener for gold smelting in China, as shown in Figure 4.

Under the flotation process, the ore slurry is concentrated by a thickener. In this work, to measure the feed concentration properly, we focused on the feed process. The slurry produced by the flotation process is merged into the slurry pump pool and then discharged from the slurry pump to the thickener for the thickening process. The slurry pipeline is equipped with a flow meter. The velocity of the feed slurry discharged into the thickener can be detected online but not the density. The feed slurry enters the thickener for settlement, and the slurry flows out from the bottom discharge port into the dehydration process. It is impossible to calculate the real-time output and cumulative output of the slurry, which makes it difficult to achieve optimal control of the production process. Because the feed density in the thickener is currently the key index for the thickening process, this study focused on the online measurement of the feed density. The process flow chart from flotation to thickening is shown in Figure 5.

## 4. Intelligent Model Based on HGA-LSTM

The intelligent model modeling process based on the LSTM network is shown in Figure 6.

### 4.1. Data Preprocessing

The data were measured using local detection devices, which have gross errors and random errors. Therefore, raw data cleaning was necessary.

The 3σ criterion was used to eliminate abnormal data in this study, as follows:

The sample was set as $x_{1}$, $x_{2}$,..., $x_{n}$, and then the 3σ of the sample was calculated according to the following formulas.
(12)$$\bar{x}=\frac{1}{n}\sum_{i=1}^{n}x_{i}$$
(13)$$\sigma =\sqrt{\frac{1}{n-1}\sum_{i=1}^{n}{\left(x_{i}-\bar{x}\right)}^{2}}$$

When data $x_{d}\left(1\leq d\leq n\right)$ satisfies Equation (13), the data are considered abnormal data or error data and should be removed.
(14)$$\left|x_{d}-\bar{x}\right|>3\sigma$$

In addition, the normalized and de-normalization equations are expressed as:(15)$$X_{i}=\frac{x_{i}-x_{min}}{x_{max}-x_{min}}$$
(16)$$x_{i}=X_{i}\times (x_{max}-x_{min})+x_{min}$$

$x_{\max}$ and $x_{min}$ represent the maximum and minimum values of the sample set, $X_{i}$ represents the normalized value, and $x_{i}$ represents the de-normalized value.

### 4.2. LSTM Network Training and Hyperparameter Optimization

As mentioned above, hyperparameter optimization, including the time window and network structure parameters, can affect the results of the LSTM network. Therefore, this work adopted an HGA-LSTM network model. Usually, neural networks with better structures have advantages in updating weights, which also may lead to additional calculations and longer training and testing times. Thus, the structure parameters of the neural network must be suitable for the training set. In addition, because the LSTM network can make good use of the past time in the training process, selecting an appropriate sliding time window size results in a vast difference in the performance of the network. A window with a small size causes the model to ignore significant information, and that with a large size overuses the data during training. As a result, to achieve better performance of the intelligent model, it is necessary to identify the best parameters, especially the time window and the network structure.

The learning process of the HGA-LSTM algorithm has two main stages. In the first stage, the learning involved designing and selecting reasonable LSTM network parameters. Using the HGA method, the time window size, number of units per hidden layer, and number of hidden layers are calculated. Two activation functions are commonly used in the LSTM model: the tanh function, which is utilized as a state activation function of the input nodes and hidden nodes, and the sigmoid function is used for the gates. To improve the generalization ability of the LSTM model, dropout is necessary to effectively reduce data overfitting. Furthermore, a gradient-based “Adam” optimizer adjusts the initialized random weight of the network, which is appropriate for problems with large-scale parameters and data.

In the second stage, to evaluate the fitness of the HGA strategy, various optimization parameters were utilized. First, the population of chromosomes with a feasible solution was initialized with random values. Each chromosome[N] contains the hyperparameters in LSTM, i.e., [N] = [time windows, number of LSTM hidden-layers, number of fully connected hidden-layers, number of units per hidden layer].

In addition, the initialized chromosomes were encoded in binary bits in this study, which represented the time window size, the number of hidden layers, and the number of units per hidden layer. Based on the selection, crossover, and mutation, the solution space was constantly searched to identify the optimal solution. In the fitness function, the performance of this model is evaluated by the root mean square error (*RMSE*) and the average relative error (*ARGE*). The *RMSE* and *ARGE* were calculated using the following formulas:(17)$$ARGE=\frac{\sum_{i=1}^{N}\left|\frac{y_{i}-pred_{i}}{y_{i}}\right|}{N}$$
(18)$$RMSE=\sqrt{\frac{\sum_{i=1}^{N}{\left(y_{i}-pred_{i}\right)}^{2}}{N}}$$

For population selection, this study used both the roulette wheel selection and the elitism policy such that chromosomes with higher fitness values had a higher probability of being selected, and the best chromosome in the current chromosome could always be selected. Meanwhile, the SQP algorithm was utilized to quadratically optimize the convergence value of the GA; that is, the set convergence value was used as the SQP initial value, and a new fitness function was fitted such that the SQP could be used for an accurate search to achieve a more accurate convergence. When selecting additional decision vectors of the GA to fit a new fitness function, bad initial vectors may be introduced into the result, and when fewer decision vectors are selected, the fitness function cannot be fitted more realistically. Therefore, we selected five decision vectors to fit the SQP fitness function through trial experiments. Finally, we discretized the decision vector obtained through the optimization of the SQP algorithm.
(19)$$\left\{ \begin{array}{l} \min fitness(x)\\ subject\;to\;g_{i}(x)\leq 0\;(i=1,2,\ldots ,m_{p})\\ \;\;\;\;\;\;\;\;\;\;\;\;\;\;\;\;\;\;g_{i}(x)=0\;(i=m_{p}+1,\ldots ,m) \end{array}\right.$$

fitness(x) represents the objective optimization fitted by the decision vector, that is, the *RMSE* of the LSTM test set, and $g_{i}(x)$ represents the boundary conditions or the limitation of the LSTM network structure and time window. The HGA-LSTM flow diagram is shown in Figure 7.

In Algorithm 1, the pseudo-code of the HGA-LSTM is shown.
**Algorithm 1** HGA–LSTM steps1. Divide the raw data into a test set and training set;    2. Use the test data to evaluate the LSTM;    3. Set GA parameters, and initialize population p randomly;    4. Select the *RMSE* of LSTM in the testing set as the fitness function of GA;    5. While the prediction criteria are not satisfied:       (a) Select befitting parents from the population;       (b) Generate a new population through crossover and mutation of chromosomes;       (c) Consider the individual chromosome that includes the time window, hidden layers, and number of hidden units per hidden layer into the LSTM to evaluate the fitness of the new population;       End 6. Set the five fast convergence values $x_{0}$ of output GA as the initial values of SQP, fit a modified fitness function, and set k=0;7. Calculate the quasi-Newton approximation matrix $H_{k}$ of the language function using the BFGS method at $x_{k}$;8. Calculate the search direction $d_{k}$, and select the appropriate step length parameter $a_{k}$;If satisfactory, stop; else set $x_{k+1}=x_{k}+a_{k}d_{k}$, k=k+1, and return to step 7;9. Discrete and output the optimal solution of the SQP, which is the hyperparameter of the LSTM network; 10. Use the well-trained LSTM network for soft sensor modeling, and evaluate the predicted results.

## 5. Experimental Results

### 5.1. Dataset Description

In industrial applications, when the nature of the conveying feed slurry is stable, the pump runs stably, the power frequency of the pump motor is fixed, and the pipeline characteristics are stable, there is a corresponding relationship between the feed density and the current, frequency, and velocity of flow [35]. Thus, the three related indicators were selected as the auxiliary variable set and are listed in Table 1.

Raw data is collected at a sampling rate of one high-quality sample every 10 s in 24 h from an actual thickener working in China. There were 8640 samples in the dataset, including one target variable and three process variables. The time series data of the variables listed in Table 1 were selected as the input, and the feed density in the dataset served as the output. To develop the intelligent model, the initial 80% of the data is used to train the network; 20% of the data is used to test the performance of the method.

### 5.2. Results Analysis

The development work of the intelligent model was conducted on a computing server with an Intel(R) Xeon(R) CPU E5-2620 v4 @2.10 GHz (two processors) and NVIDIA GeForce RTX 2080 Ti. The software environment was Windows Server 2019, Tensorflow-gpu 1.14.0, Python 3.7, and Keras 2.3.1. Taking full advantage of GPU computing, the LSTM training time for each epoch was less than 2mathrm{~s] with a total of 30 epochs. In view of the proposed HGA-LSTM model, a simulation on the abovementioned dataset was conducted. The experimental results and analysis are presented as follows: first, the output of the hyperparameter optimization from the HGA process is illustrated. Then, the prediction results of the HGA-LSTM are provided. Finally, we compare the performance of the proposed model with other models.

In the experiment, the initial population size of the GA was set to 30, with a mutation rate of 0.1, a crossover rate of 50%, and the generations set to 20 as the stop condition. For the feed density prediction, the structure of the LSTM network and time windows were optimized using the GA and HGA, respectively. In both the GA and HGA, the hidden layer contains a fully connected layer and two LSTM layers. The other parameter configurations of the HGA-LSTM and GA-LSTM were consistent with those of the LSTM. In Table 2, the specific optimization outcomes are listed.

In addition, a dropout is necessary for reducing data overfitting effectively, and thus, it improves the model generalization ability. Selecting a befitting dropout rate is crucial because if the dropout probability is very low, it leads to an underfitting phenomenon, while an excessively high dropout probability loses the benefits of adding layers. Therefore, when the accuracy of the predicted outcome does not reach the required value, the outcome is refreshed continuously by updating the dropout to ensure the required prediction accuracy. The results are presented in Figure 8.

When the dropout probability reached 0.2, the *RMSE* of the predicted value of the feed density in the thickener was approximately 3.08, and there was a good match between the actual and predicted data. However, when the dropout probability increased or decreased, the *RMSE* significantly increased, and the mismatch became significant. In summary, the dropout probability was optimized in a trial-and-error manner using feedback from the predictive data. Finally, the model results of the feed ore density based on the HGA-LSTM, GA-LSTM, and LSTM were obtained, as presented in Figure 9.

The LSTM model was optimized through trial and error based on heuristics using the feedback of the predicted data. The performance of the LSTM, GA-LSTM, and HGA-LSTM on the testing dataset is presented in Table 3.

As shown in Table 3, the HGA-LSTM model is significantly better than the GA-LSTM and LSTM models. The predicted *RMSE* values of the LSTM, GA-LSTM, and HGA-LSTM models were 3.83, 3.21, and 3.08, respectively, and the predicted result was enhanced by 15.45% and 19.5% compared to those of the GA-LSTM and LSTM models, respectively. The predicted *ARGE* values of the GA-LSTM and HGA-LSTM models were 0.0839 and 0.0752, respectively, and the predicted outcome was enhanced by 26.5% and 36.8% compared to those of the GA-LSTM and LSTM models, respectively.

The outstanding performance derived from the HGA-LSTM model was probably because the architecture of the LSTM network and time window were optimized effectively by combining the excellent global and local searching capabilities. The results demonstrate that the proper adjustment of parameters plays a critical role in achieving the desired performance. Thus, the highly effective proposed method can be used to ascertain the optimal hyperparameters for intelligent models based on deep learning algorithms, and this work expresses the potential for its application in actual industrial cases.

## 6. Conclusions

In this study, an efficient and potentially intelligent model was proposed to determine the feed ore density in a thickener. The main strategy was to employ an LSTM function optimized by an HGA combining the SQP algorithm and GA. In this method, the HGA was used to search the appropriate hyperparameters of the LSTM to improve the modeling performance. Finally, the proposed intelligent model based on the HGA-LSTM was successfully applied to an actual thickener case. This work can be extended to other scenarios where online measurement of pulp concentration is required.

## Figures and Tables

**Figure 1 materials-15-07586-f001:**
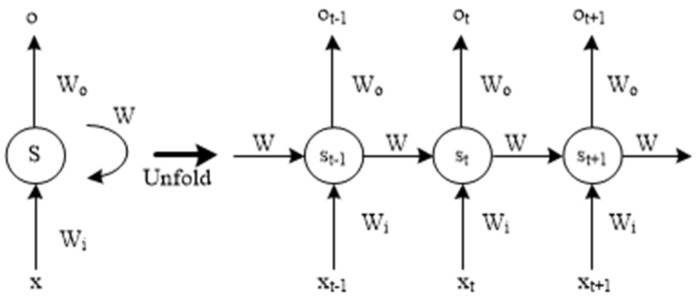
The simplified structure of RNN.

**Figure 2 materials-15-07586-f002:**
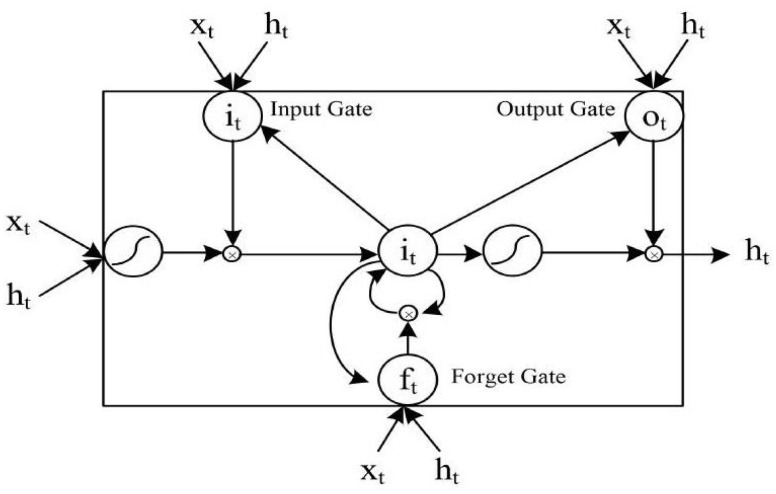
Internal structure of LSTM.

**Figure 3 materials-15-07586-f003:**
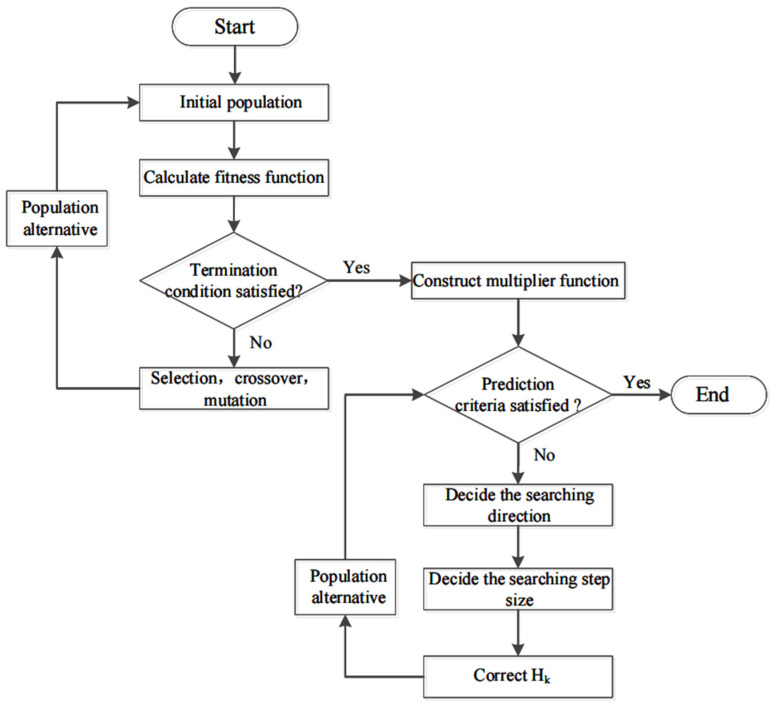
Flowchart of hybrid genetic algorithm.

**Figure 4 materials-15-07586-f004:**
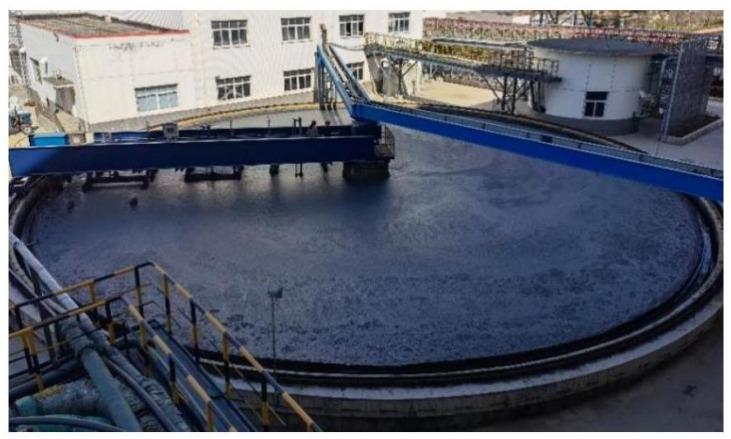
Layout of the thickener.

**Figure 5 materials-15-07586-f005:**
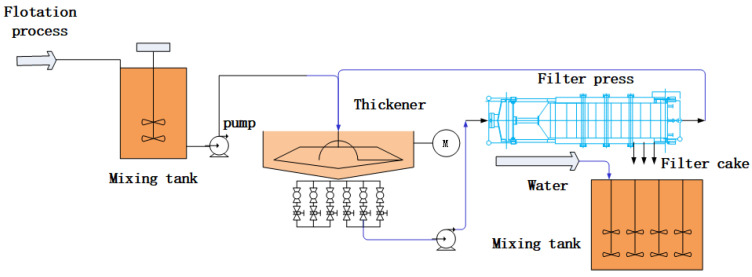
Process flow chart from flotation to thickening.

**Figure 6 materials-15-07586-f006:**
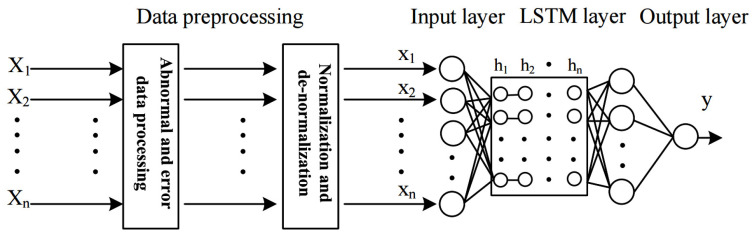
Modeling process of the intelligent model.

**Figure 7 materials-15-07586-f007:**
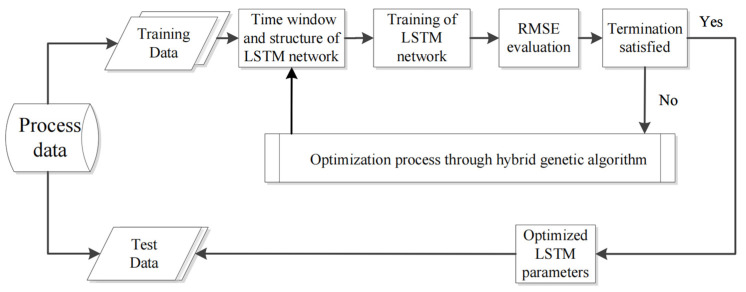
Flow diagram of the HGA-LSTM algorithm.

**Figure 8 materials-15-07586-f008:**
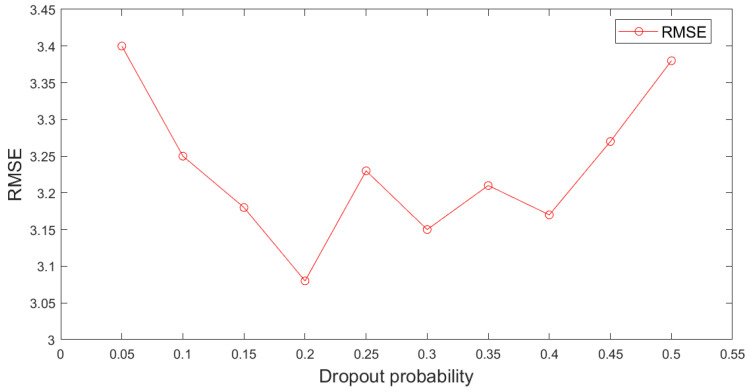
Auxiliary variables used in the intelligent model.

**Figure 9 materials-15-07586-f009:**
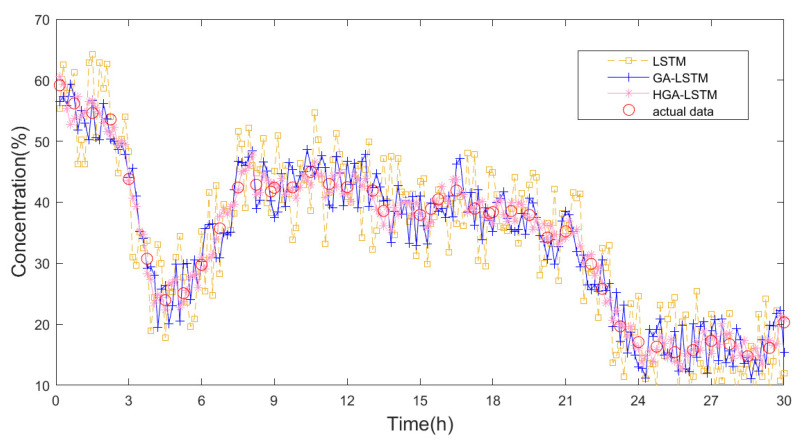
The predicted value of LSTM, GA-LSTM, and HGA-LSTM.

**Table 1 materials-15-07586-t001:** Auxiliary variables used in the soft sensing models.

Variable	Description
$x_{1}$	current of the pump
$x_{2}$	frequency of the pump
$x_{3}$	flow rate of the pump

**Table 2 materials-15-07586-t002:** Specific optimization outcomes based on GA and HGA.

Parameter	GA-LSTM	HGA-LSTM	LSTM
Time windows	11	11	10
Number of LSTM hidden-layers	2	2	2
Number of fully connected hidden-layers	1	1	1
LSTM units on the first layer	94	88	90
LSTM units on the second layer	55	53	60
Fully connected units on the third layer	70	72	80

**Table 3 materials-15-07586-t003:** Comparison results of LSTM, GA-LSTM, and HGA-LSTM.

Method	*RMSE*	Improvement (%)	*ARGE*	Improvement (%)
LSTM	3.83	-	0.119	-
GA-LSTM	3.21	15.45	0.0839	26.5
HGA-LSTM	3.08	19.5	0.0752	36.8

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
