# Peer review of "An HGA-LSTM-Based Intelligent Model for Ore Pulp Density in the Hydrometallurgical Process"

_materials, 2022, doi:10.3390/ma15217586_

Round 1

Reviewer 1 Report

Review comments from the reviewer

Article title: An HGA-LSTM Based Intelligent Model for Ore Pulp Density in Hydrometallurgical Process

This research presents an intelligent model for predicting ore pulp density in hydrometallurgical process using LSTM and hybrid genetic algorithm. It is noted that the proposed intelligent method is applied to a real thickener case in China. The objective and quality of the work is appreciable and has merit to publish by the journal. I have a few comments below that may help the authors to improve their work:

Review comments from the reviewer

  1. In the introduction section the existing research gap is not explained.
  2. The drawbacks of the machine learning models applied for the proposed problems need to be highlighted in the introduction section.
  3. It is necessary to reason out the choice of LSTM for this problem statement.  
  4. The backend math involved in the internal working of the LSTM need to be explained.
  5. The need and operation of the language function need to be explained in equation 9.
  6. Equation 10 doesn’t seem to be meaningful. Please check. It should be restructured.
  7. An overall figure explains the entire proposed work should be included in section 3.
  8. What is the dataset used for building the intelligent model?
  9. Dataset description should be included explaining all the available features.
  10. The reason for doing data preprocessing should be explained.
  11. What proportion of training data and testing data used in the proposed work?
  12. Separate algorithm explaining the working of LSTM and HGA should be included.
  13. The combination part where the LSTM and HGA are integrated should be explained in detail.
  14. Section 5 need to be reorganized. Why data collection and variable section are included in experimental results?
  15. What is mean by Auxiliary variable? What is need for it? How it is selected?
  16. All the images in Figure 8 and Figure 9 need to be updated. No inferences can be derived form that images.  
  17. Many possible other parameters are available for performance comparison. All the possible parameters should be considered and compared.
  18. Overall English need to be verified and enhanced.
  19. Few additional references mentioned below related to LSTM can be included. The contents should be referred and included for better understanding.

https://doi.org/10.1016/j.jweia.2021.104820.

 https://doi.org/10.3390/s21072515

https://doi.org/10.1007/s00521-021-05690-8

Author Response

       We feel great thanks for your professional review work on our paper. As you are concerned, there are several problems that need to be addressed. According to your nice suggestions, we have made extensive correction to our previous draft. We look forward to hearing from you regarding our submission. We would be glad to response to any further questions and comments that you may have.

  1. In the introduction section the existing research gap is not explained.

Response: First of all, thank you for your suggestion. The description of research gap is added in the introduction. Optimal control of the thickening process usually depends on the quality variables, such as feed density, which is difficult to online measure. Because the density of the feeding ore usually fluctuates substantially and existence of nonlinearity. However, there is no research on the application of real-time online measurement method of thickener feed pulp density in the actual production process.

  1. The drawbacks of the machine learning models applied for the proposed problems need to be highlighted in the introduction section.

Response: According to your comment, we have added relevant content in the introduction.

  1. It is necessary to reason out the choice of LSTM for this problem statement.

Response: Thank you for your instructive comment. We have re-written this part according to your suggestion.

Industrial processes are naturally dynamic, and the data series are sampled in real time from a continuous process. Thus, to model such data sequences more accurately, the dynamic characteristics must also be considered; that is, the models must utilize past states and information to predict the present state. Recurrent neural networks (RNNs) is a dynamic neural network. It can suffer from gradient explosion and gradient disappearance, because the memory function of past information. Thus, a long short-term memory (LSTM) network which add gate units to retain short and long-term memories is poposed to deal with this problem.

  1. The backend math involved in the internal working of the LSTM need to be explained.

Response: Special thanks to you for your good comments. We have added relevant content. Due to the pages limit, more backend math involved LSTM can be found in reference[21].

  1. The need and operation of the language function need to be explained in equation 9.

Response: Special thanks to you for your good comments. We have added relevant content, but Due to the limited number of pages, more information about language function can be found in reference[33].

  1. Equation 10 doesn’t seem to be meaningful. Please check. It should be restructured.

Response: Thank you for your suggestion. We have checked and restructured the Equation 10.

  1. An overall figure explains the entire proposed work should be included in section 3.

Response: Thank you for your comment. Figure7 explains the entire proposed work, which is placed on page 8. We have added relevant content about this figure.

  1. What is the dataset used for building the intelligent model?

Response: Thank you for your comment. The raw data which at a sampling rate of one high-quality sample every 10 s in 24h is collected from an actual thickener working in China. There were 8640 samples in the dataset, including one target variable and three process variables. The time series data of the variables listed in Table1 were selected as the input, and the feed density in the dataset served as the output.

  1. Dataset description should be included explaining all the available features.

Response: Thank you for your comment. There were 8640 samples in the dataset, including one target variable and three process variables. The time series data of the variables listed in Table1 were selected as the input, and the feed density in the dataset served as the output. To develop the intelligent model, the initial 80% of the data is used to train the network; the 20% of the data is used to test the performance of the method.

  1. The reason for doing data preprocessing should be explained.

Response: Thank you for your comment. Because raw data comes from the actual production process and there are many noises and singularities of data, we must process the raw data before training the model.

  1. What proportion of training data and testing data used in the proposed work?

Response: Thank you for your comment. To develop the intelligent model, the initial 80% of the data is used to train the network; the 20% of the data is used to test the performance of the method.

  1. Separate algorithm explaining the working of LSTM and HGA should be included.

Response: Thank you for your comment. In section 2, we have revised the content to explain the working of LSTM and HGA.

  1. The combination part where the LSTM and HGA are integrated should be explained in detail.

Response: Thank you for your comment. In subsection 4.2, We have added relevant content to explain the combination part of LSTM and HGA.

  1. Section 5 need to be reorganized. Why data collection and variable section are included in experimental results?

Response: Thank you for your suggestion. We have reorganized the Section 5.

  1. What is mean by Auxiliary variable? What is need for it? How it is selected?

Response: Thank you for your suggestion. When training the model, we select the most relevant input data according to the process mechanism and production experience. We have added references[35] to illustrate this issue.

  1. All the images in Figure 8 and Figure 9 need to be updated. No inferences can be derived form that images.

Response: Thank you for your comment. We have updated the Figure 8 and Figure 9, and revised relevant content about Figure 8 and Figure 9. Please see the attachment.

  1. Many possible other parameters are available for performance comparison. All the possible parameters should be considered and compared.

Response: Thank you for your professional comment. In the fitness function, the performance of the model is evaluated by the root mean square error (RMSE) and the average relative error (ARGE).It is enough to compare the advantages and disadvantages of the corresponding methods for the problems in this research. We plan to use more comparison methods to evaluate the model performance in the further research based on this article, and look forward to your professional suggestions in the next time. Thank you.

  1. Overall English need to be verified and enhanced.

Response: Thank you for your suggestion. We carefully revised our English and asked my international student friends for advice. The English level of this article has been enhanced. Thank you very much.

  1. Few additional references mentioned below related to LSTM can be included. The contents should be referred and included for better understanding.

Response: We add some references that can improve this research. Thank you for your good suggestion.

[6]Predicting wind flow around buildings using deep learning.Journal of Wind Engineering and Industrial Aerodynamics 2021, 219, 104820.

[7]Wind-Induced Pressure Prediction on Tall Buildings Using Generative Adversarial Imputation Network. Sensors 2021, 21.

[9]Surface crack detection using deep learning with shallow CNN architecture for enhanced computation. Neural Computing and Applications 2021, 33.

Reviewer 2 Report

The application of Long Short-Term Memory (LSTM) in different fields has become very widespread, the main problem being the consideration of the limitations imposed on each field.

The authors of the research implemented an intelligent model for the ore pulp density in hydrometallurgical process.

General observations:

-      the large number of bibliographic references and the correct way of structuring the current state in the studied field are positively noted (29 out of a total of 34 references being used in the introduction);

-   please specify the meaning of LSTM - long short-term memory (appears in the abstract);

-         please try to recreate the process flow diagram from figure 5; the whole article shows professionalism, but this figure seems to have been made by a child, use standardized symbols (you can find them in the Autodesk library, for example). Example: I don't understand what you wanted to draw to identify the Slurry pump pool;

-         line 201 - I don't think equation 17 is well identified in the following sentence: If a certain sample data xd(1 ≤ d ≤ n) satisfies Equation (17), the data are considered as abnormal data or error data and should be removed;

-         abbreviations should be explained before they appear (Example: RMSE appears in figure 7 and is explained on line 238);

Statistically, over 75% of researches do not translate the research results into industry practice. In order for the authors not to be part of this statistic, please extract, from the research carried out, a practical conclusion that can be applied by specialists in the field and that will lead to the improvement of a technological process.

Author Response

      We feel great thanks for your professional review work on our paper. As you are concerned, there are several problems that need to be addressed. According to your nice suggestions, we have made extensive correction to our previous draft. We look forward to hearing from you regarding our submission. We would be glad to response to any further questions and comments that you may have.

  1. the large number of bibliographic references and the correct way of structuring the current state in the studied field are positively noted (29 out of a total of 34 references being used in the introduction);

Response: Thank you for your comment.

  1. please specify the meaning of LSTM - long short-term memory (appears in the abstract);

Response: Thank you for your comment. We have corrected this problem in the abstract.

  1. please try to recreate the process flow diagram from figure 5; the whole article shows professionalism, but this figure seems to have been made by a child, use standardized symbols (you can find them in the Autodesk library, for example). Example: I don't understand what you wanted to draw to identify the Slurry pump pool;

Response: Thank you for your good suggestion. We have redrawn figure 5 and hope it will be useful to you and other readers. Please see the attachment.

  1. line 201. I don't think equation 17 is well identified in the following sentence: If a certain sample data xd(1≤d≤n) satisfies Equation (17), the data are considered as abnormal data or error data and should be removed;

Response: We have corrected this error. Thank you very much for your detailed comments.

  1. abbreviations should be explained before they appear (Example: RMSE appears in figure 7 and is explained on line 238);

Response: We have revised it according to your good comment. Thank you very much.

  1. Statistically, over 75% of researches do not translate the research results into industry practice. In order for the authors not to be part of this statistic, please extract, from the research carried out, a practical conclusion that can be applied by specialists in the field and that will lead to the improvement of a technological process.

Response: This research is based on the needs of the actual process. We are honored to apply our research to the actual process. For scenes where the slurry is transported by pump, the intelligent model can be tried to measure the slurry concentration. I hope it can help other researchers. We have added relevant content to the conclusion. Thank you very much.
